# MicroRNAs in Myocarditis—Review of the Preclinical In Vivo Trials

**DOI:** 10.3390/biomedicines11102723

**Published:** 2023-10-08

**Authors:** Grzegorz Procyk, Olga Grodzka, Marcelina Procyk, Aleksandra Gąsecka, Katarzyna Głuszek, Małgorzata Wrzosek

**Affiliations:** 11st Chair and Department of Cardiology, Medical University of Warsaw, Banacha 1A, 02-097 Warsaw, Poland; 2Doctoral School, Medical University of Warsaw, 02-091 Warsaw, Poland; 3Department of Neurology, Faculty of Medicine and Dentistry, Medical University of Warsaw, Ceglowska 80, 01-809 Warsaw, Poland; 4Faculty of Biology and Biotechnology, Warsaw University of Life Sciences (WULS-SGGW), 02-787 Warsaw, Poland; 5Collegium Medicum, Jan Kochanowski University of Kielce, 25-406 Kielce, Poland; 6Department of Biochemistry and Pharmacogenomics, Medical University of Warsaw, Banacha 1, 02-097 Warsaw, Poland

**Keywords:** myocarditis, microRNA, inflammation, preclinical trials, in vivo trials

## Abstract

Myocarditis is an inflammatory heart disease with viruses as the most common cause. Regardless of multiple studies that have recently been conducted, the diagnostic options still need to be improved. Although endomyocardial biopsy is known as a diagnostic gold standard, it is invasive and, thus, only sometimes performed. Novel techniques of cardiac magnetic resonance are not readily available. Therapy in viral infections is based mainly on symptomatic treatment, while steroids and intravenous immunoglobulins are used in autoimmune myocarditis. The effectiveness of neither of these methods has been explicitly proven to date. Therefore, novel diagnostic and therapeutic strategies are highly needed. MiRNAs are small, non-coding molecules that regulate fundamental cell functions, including differentiation, metabolism, and apoptosis. They present altered levels in different diseases, including myocarditis. Numerous studies investigating the role of miRNAs in myocarditis have already been conducted. In this review, we discussed only the original preclinical in vivo research. We eventually included 30 studies relevant to the discussed area. The altered miRNA levels have been observed, including upregulation and downregulation of different miRNAs in the mice models of myocarditis. Furthermore, the administration of mimics or inhibitors of particular miRNAs was shown to significantly influence inflammation, morphology, and function of the heart and overall survival. Finally, some studies presented prospective advantages in vaccine development.

## 1. Introduction

### 1.1. Myocarditis—Definition, Etiology, and Epidemiology

Myocarditis is an inflammatory, polymorphic disease of the heart [1]. It is one of the most challenging cardiac disorders partially due to diverse etiology and complex pathophysiology. It may be caused by viruses (most commonly), bacteria, or autoimmunization [2]. Importantly, it can also be caused by medicinal drugs, such as antipsychotics [3]. All these factors are responsible for the cardiac damage that initiates an immune response. Severe inflammation causes further damage to cardiomyocytes, and in the worst-case scenario, it can lead to non-ischemic heart failure [4]. The percentage of heart failure cases in patients suffering from myocarditis was estimated at 0.5–4% [5].

At the molecular level, the inflammation in myocarditis is mediated via the cytokine interleukin-1 (IL-1), among others. Therefore, it seems a promising therapeutic option for patients with myocarditis to inhibit IL-1 with the use of monoclonal antibodies, such as anakinra [6].

This potentially life-threatening disease usually affects young people without any severe cardiovascular risk factors [7] and is a significant cause of sudden unexpected death (SUD) in young athletes [8]. In a population-based autopsy study by Li et al., SUDs due to myocarditis accounted for 0.17% of all SUDs and 0.70% of autopsied SUDs. The authors found that the most common histologic types of myocarditis were, respectively, lymphocytic (54.4%), neutrophilic (31.1%), eosinophilic (12.6%), and giant cell type (1.9%) [9].

### 1.2. Myocarditis—Diagnosis and Treatment

Myocarditis presents a large variety in etiology, pathophysiology, and clinical manifestation. Hence, it is a significant challenge in diagnosis and differentiation from other cardiac diseases [10]. Currently, an endomyocardial biopsy remains a gold standard diagnostic test for myocarditis. However, it is a highly invasive method, and thus it is only sometimes performed [11]. Cardiac magnetic resonance (CMR) possesses the ability of multiparametric tissue characterization and is considered a non-invasive gold standard method in myocarditis diagnosis [12,13]. According to the remarkably heterogenic character of the disease, it is challenging to find sufficiently specific and sensitive biomarkers [14]. Those currently used were shown to be present in many more cardiac disorders [13,15].

Therapeutic strategies in myocarditis remain limited. Treatment for the majority of viral infections is either symptomatic or supportive. As the viruses, precisely Coxsackievirus B3 (CVB3), are the leading cause of myocarditis, there is a high demand for novel anti-viral agents [16]. Systemic steroids and intravenous injections of immunoglobulins have been used in autoimmune myocarditis [17]. Steroids showed some beneficial effects in alleviating the disease [17,18]. However, there was no evidence of an advantage in performing intravenous injections of immunoglobulin [18]. Myocarditis can also be caused iatrogenically by treatment with immune checkpoint inhibitors (ICIs). Proprotein convertase subtilisin/kexin type 9 inhibitor (PCSK9i) was shown to reduce atherosclerotic cardiovascular events in cancer patients treated with ICIs, and thus, it may also potentially help patients with myocarditis caused by ICIs [19]. Some data suggest a potential cardioprotective effect of nutraceuticals: quercetin, polydatin, and omega-3 fatty acids (e.g., eicosapentaenoic acid and docosahexaenoic acid) in myocarditis. More precisely-quercetin was shown to increase cell viability and alleviate the release of cytokines. Similarly, polydatin was found to reduce cytokine storms in cardiomyocytes, while omega-3 fatty acids were correlated with both decreased expression of inflammatory cells and suppressed levels of nuclear factor kappa-light-chain-enhancer of activated B cells (NF-κB) [20,21,22]. Nevertheless, effective treatment for myocarditis is still lacking [23].

### 1.3. MicroRNAs—Synthesis, Function, and Role in Myocarditis

MicroRNA (miRNAs/miRs) are a class of short non-coding regulatory ribonucleic acids (RNAs) [24]. They inhibit the translation by binding to the target mRNA (messenger RNA) sequences [25]. In primary, miRNAs are initially derived from co-transcription with the approximal genes or within the miRNA cluster [26]. Subsequently, the following modifications occur the first by the enzyme Drosha into pre-miRNAs and the second by the enzyme Dicer into mature miRNAs [27]. The last step is to load the miRNA into the Argonaut 2 protein as a part of the RNA-induced silencing complex (RISC). RISC can inhibit the complementary mRNA by binding the 3′ end of silenced mRNA to the 5′ end of the miRNA [28]. Moreover, there are other mechanisms of miRNAs’ production, independent of Drosha and Dicer with noncanonical binding sides. miRNAs are resistant to RNAse activity and remain stable in circulation, making them promising novel biomarkers [29].

The role of miRNAs in biological functions has been extensively studied. MiRNAs are included in cell differentiation, metabolism, and apoptosis [30]. They have been studied in various conditions, including many cardiovascular diseases [31,32,33]. Characteristic patterns of circulating miRNAs have been observed in different cardiovascular diseases, including myocarditis [34]. They play an important role in myocarditis pathogenesis and thus can be used as diagnostic and therapeutic molecules [35]. Interfering with the miRNAs has been demonstrated to reduce myocardial damage in both viral and autoimmune in vivo models [5]. It was reported in much research that miRNAs play a crucial role in viral replication. However, they are also susceptible to being regulated by viruses [36].

According to all the above information, the putative role of miRNAs in diagnosing and treating myocarditis is suggested. Although no precise conclusions have been drawn to date, and there is much more to investigate, there are undoubtedly many vital issues to be discussed based on the already conducted studies. We have previously summarized the current evidence about the role of miRNAs in myocarditis based on clinical studies [37]. This review further discusses this issue, outlining available data from preclinical in vivo studies.

## 2. MicroRNAs in an In Vivo Model of Myocarditis

### 2.1. Methodology

Numerous studies investigating the role of miRNAs in myocarditis have already been conducted. In this review, we included and discussed only original preclinical in vivo research. Reviews, letters to the editors, and commentaries were not included. Clinical and preclinical in vitro studies were excluded. We searched the PubMed Database by the query: “(miRNA OR microRNA) AND (myocarditis)”. Excluding unsuitable titles, article types, or abstracts, we retrieved in complete form and assessed the remaining studies. We eventually included 30 original preclinical in vivo research relevant to the discussed area. We divided included research into three paragraphs: (i) dysregulated miRNAs in myocarditis (in heart tissues and circulating), (ii) intervention in viral myocarditis (VMC) in vivo models, and (iii) intervention in autoimmune myocarditis (AMC) in vivo models. One of the studies was relevant for both Section 2.1 and Section 2.2 and thus was adequately described in these parts (Figure 1). The main limitation of the used methodology is searching solely PubMed Database, which might have resulted in missing papers not indexed in this database. Nevertheless, it yielded primarily 184 records, which allowed us to analyze the topic thoroughly and draw crucial conclusions.

### 2.2. Dysregulated miRNAs

The dysregulation of miRNAs has been observed in many preclinical studies, including in vivo models of myocarditis, most commonly the VMC model. Analyzed research has shown upregulation or downregulation of particular miRNAs. Some studies focused on measuring multiple miRNA levels, while others considered only one or several molecules. The dysregulation was demonstrated either in circulating miRNAs or in heart tissue miRNAs.

#### 2.2.1. miRNAs Dysregulated in Heart Tissue

The significant dysregulation of miRNA levels in heart tissue was found in the studies discussed below. Wu et al. analyzed the miRNA levels in Coxsackievirus A2-infected mice, comparing them to non-treated controls. Deregulation of 52 miRNAs at 3. days post-infection and 73 miRNAs at 7 days post-infection was observed in the group of infected mice. At the same time, no significant changes were found in the control group [38]. Wang et al. assessed the levels of different miRNAs in cardiomyocytes derived from mice affected by VMC. In microarray analysis, they showed a downregulation of 67 and an upregulation of 27 other miRNAs, whereas the polymerase chain reaction (PCR) demonstrated elevated levels of 7 and lowered levels of 18 miRNAs. These 25 miRNAs, assessed as altered in PCR, can be potentially included in novel prognostic assays and therapeutic approaches in VMC [39]. Similar research was conducted by Corsten et al., who studied the levels of multiple miRNAs in mice. The mice infected with CVB3, either susceptible to myocarditis or resistant, were compared to non-infected mice. The deregulation of 221 different miRNAs was shown in the susceptible group, while in the resistant group, only 78 miRNAs had altered expression levels [40]. Other studies demonstrated altered levels of particular miRNAs, thus implicating their prospective role as diagnostic biomarkers in myocarditis. Hemida et al. studied the levels of miR-208 in CVB3-infected mice. It was observed that mice with myocarditis showed higher levels of miR-208 than healthy controls. The zinc finger protein-48 (ZFP-148), crucial in viral replication, was identified as a target for miR-208 [41]. Lin et al. analyzed the role of miR-19b in VMC. It was shown that the level of miR-19b was significantly higher in VMC mice compared to healthy individuals [42]. Xu et al. compared the expression levels of miR-1 between mice infected with CVB3 and healthy individuals. The first group presented higher levels of this miRNA than the latter [43]. Zhang et al. also studied the role of miRNAs in mice affected by VMC. The levels of different miRNAs were measured, and five appeared to be significantly altered. The upregulation of miR-21, miR-29a, miR-146a, miR-374, and downregulation of miR-23a was observed in mice with myocarditis compared to healthy individuals [44].

#### 2.2.2. Dysregulation of Circulating miRNAs

MiRNAs were shown to be dysregulated in blood samples derived from examined mice, which may be crucial for potential novel non-invasive diagnostic methods. Blanco-Domínguez et al. studied a group of mice with induced VMC or AMC and compared them to the group of mice with induced myocardial infarction. It was demonstrated that mmu-miR-721 (synthesized by Th17 cells) was present in the plasma of mice with myocarditis while absent in the plasma of mice with myocardial infarction [45]. Two other studies included analyses of both circulating and heart tissue miRNAs. Xia et al. demonstrated that miR-217 and miR-543 levels were significantly higher in CVB3-infected mice than in non-infected ones [46]. Zhang et al. conducted a study on mice with myocarditis, which were shown to have lower levels of miR-381 than healthy controls [47]. All studies discussed in this subsection with additional data are summarized in Table 1.

### 2.3. Intervention in Viral Myocarditis In Vivo Models

The administration of either a mimic or an inhibitor of a particular miRNA has been studied in much research. After the intervention, the effect exerted on several parameters, such as inflammation level, cardiac function, heart morphology, and survival rate, was assessed. Alteration of miRNA levels alleviates or exacerbates the disease’s natural course, which might be pivotal for managing myocarditis.

#### 2.3.1. Intervention in Viral Myocarditis In Vivo Models-Influence on Viral Load and Replication

A part of the analyzed research demonstrated that the intervention with either mimics or inhibitors of miRNAs might significantly impact different parameters, including viral replication or viral load. This finding might be substantial for the development of novel anti-viral therapies. Liu et al., in the in vivo part of their study, showed that mice with induced VMC had higher levels of miR-324-3p as compared to healthy ones. Further, mice with VMC were modified to overexpress miR-324-3p or to have silenced its expression. The group overexpressing this miRNA demonstrated lower levels of viral capsid protein 1 (a marker of viral replication) than the miRNA-silenced group. The target for miR-324-3p was identified as a protein tripartite motif containing 27 (TRIM27) involved in regulating immune response [48]. Corsten et al. studied the role of the miR-221/-222 cluster in VMC. It was observed that the downregulation of miR-221/-222 aggravated a cardiac injury in mice infected with CVB3 compared to the control group (CVB3-infected mice treated with respective scrambled controls). The study group showed increased viral replication, cardiac necrosis, infiltration of inflammatory cells, and elevated levels of interferon-gamma. It was demonstrated that the miR-221/-222 cluster acted as a protective factor in VMC [49]. He et al. observed that the level of miR-21 in mice presenting with myocarditis was significantly decreased compared to healthy mice. Injecting mice suffering from myocarditis with plasmids containing miR-21 reduced inflammation and increased the survival rate. Surprisingly, the group treated with miR-21 had higher viral load levels than control individuals [50]. Slightly opposite results were observed by He et al., who investigated mice presenting with VMC. After the injection of miR-21 (compared to the injection of non-mammal miR), the viral replication was suppressed in this in vivo model [51]. The following studies made use of constructing viruses containing miRNAs’ target sequences, which resulted in repressed viral replication in a living organism and might be a novel technique to generate attenuated viruses. Xiao et al. created the engineered CVB3 harboring miR-206 target sequence. This miRNA is specific for cardiac muscle tissue. It was demonstrated that the replication was partially inhibited after infecting mice with a virus modified in that way. It did not occur in the control group of mice infected with a virus containing non-mammal miRNA target sequences. Thus, the CVB3 harboring mammal tissue-specific miRNA targets was shown to be attenuated [52]. He et al. engineered CVB3 to contain target sequences for miRNAs important in suppressing VMC: miR-133 and miR-206. Afterward, these modified viruses were injected into mice. It caused the viral replication to be lowered. At the same time, mice survival was increased compared to the controls treated with either CVB3 containing negative control of miRNA target sequences or non-modified CVB3 [53].

#### 2.3.2. Intervention in Viral Myocarditis In Vivo Models-Other Analyzed Studies

Multiple studies have shown that the inhibition or overexpression of miRNAs influenced cardiac morphology and function, inflammation degree, and, not less significantly, overall survival. Therefore, miRNAs may play a crucial role in myocarditis supportive therapy. Li et al. prepared VMC rat models and treated them with miR-133a encapsulated in exosomes. They were compared to three other groups: (i) healthy rats, (ii) VMC rats not treated with miR-133a, and (iii) VMC rats with the silenced expression of miR-133a. The group treated with exosomal miR-133a showed lower levels of inflammatory factors than other groups. Noteworthily, the highest levels of these cytokines were observed in the miR-133a-silenced rats [54]. Liu et al. showed that miR-21 and miR-146b levels were significantly higher in mice presenting with VMC than in healthy individuals. Furthermore, the inflammation was alleviated after injecting mice suffering from myocarditis with a miR-21 or miR-146b inhibitor. More precisely, lowered levels of inflammatory cytokines, reduced numbers of Th17-cells, diminished cardiac inflammation, and decreased heart weight/body weight ratio were observed [55]. A similar beneficial influence of miR-21 silencing was demonstrated by Xue et al., who investigated mice infected with CVB3 to provoke VMC. The injection of miR-21 inhibitor resulted in improved ejection fraction assessed in echocardiography compared to other mice affected by myocarditis. Furthermore, CVB3-infected mice treated with negative control of miRNA or not treated at all had stiff and enlarged hearts. Moreover, inflammatory infiltration was visible, and prominently increased myocardial fibrosis was evidenced in these mice [56]. Li et al. demonstrated that mice affected by VMC presented lower levels of miR-1 and miR-133 than healthy mice. Subsequently, the group of mice with myocarditis was injected with miR-1/133-mimic (the mixture of miR-1 and miR-133) or with negative control. It was demonstrated that in the first group, cardiac function was improved, pathological changes in the myocardium were attenuated, and cardiomyocyte apoptosis was reduced [57]. Gou et al. showed a putative anti-inflammatory role of miR-223. Although mice infected with CVB3 demonstrated lowered levels of miR-223, the upregulation of this miRNA caused a decrease in the levels of inflammatory factors. Thus, miR-223 might protect against inflammation in VMC [58]. Li et al. demonstrated that CVB3-infected mice presented decreased levels of miR-425-3p. Moreover, after injecting agomiR-425-3p into mice with VMC, it appeared that IL-6, IL-12, and TNF-α levels were lower than in mice without this intervention. Furthermore, the survival rate of VMC mice was higher in those who had received the injection of agomiR-425-3p [59]. Three studies focused directly on miR-155, which implies its pivotal role in VMC. Surprisingly, not all the results are consistent, which indicates a high demand for further studies. Zhang et al. proved that the level of miR-155 was elevated in mice suffering from VMC compared to healthy mice. Furthermore, miR-155 silencing in CVB3-infected mice (i) attenuated myocardial inflammation, (ii) reduced mortality, and (iii) improved cardiac function [60]. A similar observation was made by Corsten et al., who further studied the role of miR-155, which had appeared to be significantly upregulated in the hearts of mice suffering from VMC in previous research. The inhibition of miR-155 in mice resulted in minor body weight loss, reduced leukocytosis, and alleviated monocyte infiltration compared to control mice [40]. However, Bao et al. obtained slightly opposite results. They assessed the role of miR-155 in mice infected with CVB3. They showed that mice transfected with miR-155 presented a higher survival rate than those treated with negative control. This study highlighted miR-155 as a potential therapeutic option in VMC [61]. All studies discussed in this subsection with additional data are summarized in Table 2.

### 2.4. Intervention in Autoimmune Myocarditis In Vivo Models

Because the second most common cause of myocarditis is autoimmunization, some research focused on the experimental model of AMC. Mirna et al. analyzed mice with induced AMC. It was evidenced that miR-21a-5p was highly expressed in all mice with myocarditis. Further, one group of mice was treated with antagomiR-21a-5p while the other one received negative control oligonucleotide. Treatment with the miR-21a-5p antagonist in the study group significantly reduced myocardial inflammation and fibrosis compared to the control group [62]. Sun et al. isolated exosomes containing miR-142 from mice with induced AMC and from healthy controls. The first group showed higher levels of this miRNA. Exosomes isolated from both groups were injected into naive mice, and the inflammation intensity was assessed. CD4^+^ cell infiltration was more significant in mice treated with exosomes derived from the group of mice with myocarditis, showing the role of miR-142 in inducing heart dysfunction. Moreover, the injection of miR-142 inhibitor was conducted, which alleviated cardiac injury [63]. Chen et al. analyzed the role of miR-223-3p enclosed in dendritic cells. miR-223-3p was chosen because mice that constituted the experimental model of AMC showed a decreased level of this miRNA. The agonist, inhibitor, or negative control of miR-223-3p (enclosed in dendritic cells) was injected into the mice with myocarditis. The results showed reduced pericardial inflammation and improved ejection fraction in the mice treated with miR-223-3p agonist compared to control ones [64]. Pan et al. demonstrated that miR-141-3p levels were significantly lower in mice with experimental AMC than in healthy controls. The myocarditis mice were then injected with agomiR-141-3p or miR-141-3p-inhibitor. It was observed that in the group treated with agomiR-141-3p, the inflammatory factors’ levels were lower. What is more interesting is that cardiac hypertrophy and heart-weight to body-weight ratio were also diminished in this group [65]. Yan et al. conducted a study on mice affected by experimental AMC. Mice with myocarditis presented elevated levels of miR-155 as compared to healthy controls. However, after inhibition of miR-155, Th-17 cell infiltration was diminished while left ventricular ejection fraction and fractional shortening were noted to have increased [66]. Zhao et al. investigated rats with induced experimental AMC. They presented decreased levels of miR-590-3p as compared to healthy rats. Moreover, the administration of miR-590-3p was observed to reduce levels of proinflammatory cytokines [67]. The studies mentioned above provide new insights essential for establishing a new treatment strategy in AMC. All studies discussed in this subsection with additional data are summarized in Table 3.

## 3. Conclusions and Future Perspectives

Much research about the role of miRNAs in myocarditis has been conducted so far. Much of it was conducted as preclinical in vivo studies, which we reviewed thoroughly in this paper.

The altered miRNA levels have been observed, including upregulation and downregulation of different miRNAs in the mice models of myocarditis. It is applied to both circulating and myocardial miRNAs. Whether this relationship is casual still needs to be studied.

The administration of mimics or inhibitors of particular miRNAs influenced viral replication, inflammation, cardiac damage degree, cardiac function, and overall survival. Antagonizing the effect exerted by a given miRNA, one can improve the myocarditis course. It leads to a decrease in viral load and a decrease in the levels of proinflammatory cytokines. Importantly, these positive effects were observed not only in the case of VMC but also in the case of AMC. Interestingly, viruses engineered with miRNA target sequences demonstrated reduced replication, which may highlight a novel method for constructing attenuated viruses. However, some studies demonstrated conflicting results, which justifies further research in the field.

MiRNAs are suggested to have potential use as specific and sensitive non-invasive diagnostic tools, which makes them ideal candidates for diagnostic biomarkers of myocarditis. Nevertheless, further studies must confirm their utility before their clinical implementation. Furthermore, administration of a particular miRNA’s inhibitor or analog might substantially improve both casual and symptomatic treatment of myocarditis. Last but not least-engineered viruses may be a milestone in vaccine development. Nevertheless, there is still much to be done. Considering all possible advantages of using miRNAs and miRNAs-based drugs, further studies are recommended, not only in myocarditis.

## Figures and Tables

**Figure 1 biomedicines-11-02723-f001:**
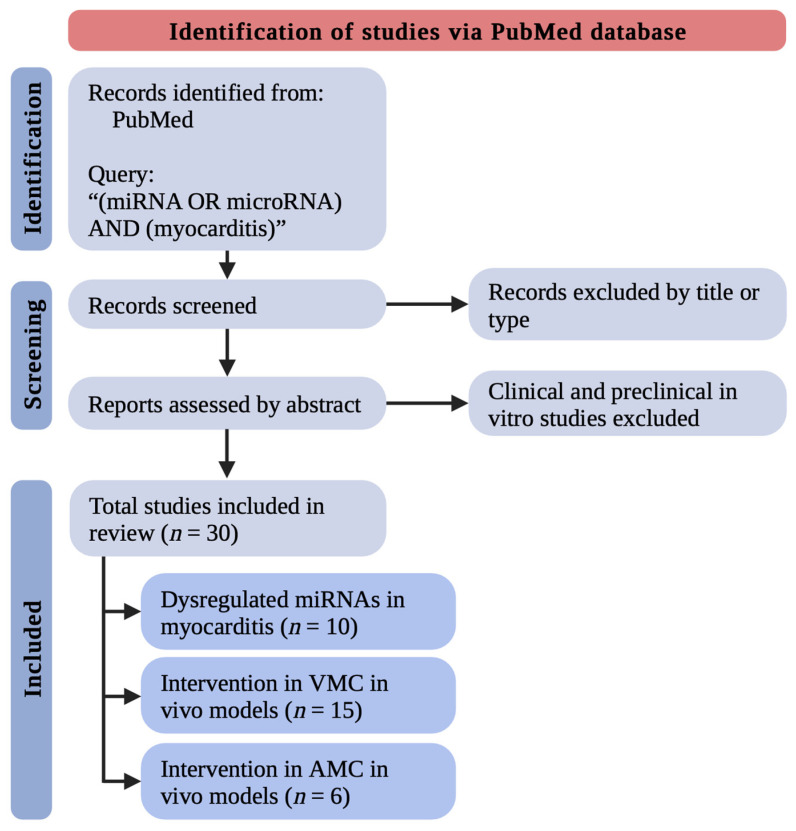
The methodology used to identify, include, and divide studies in this review. Note that studies from different parts do not sum up to 30 because one study was included in two suitable paragraphs. AMC—autoimmune myocarditis; miRNAs—micro-ribonucleic acids; *n*—a number of studies; VMC—viral myocarditis.

**Table 1 biomedicines-11-02723-t001:** Summary of recent studies regarding deregulated miRNAs in myocarditis.

Ref.	Year	Population	Comparison	miRNA	Outcome	Methodology
[38]	2022	CVA2-infected mice	non-treated mice	miRNAome	52 dysregulated miRNAs at 3 days post-infection 73 dysregulated miRNAs at 7 days post-infection	miRs in heart tissue by RNA-sequencing and qPCR
[39]	2018	5 VMC mice	5 healthy mice	miRNAome	↑ 27 miRs and ↓ 67 miRs in VMC mice assessed by microarray ↑ 18 miRs and ↓ 7 miRs in VMC mice assessed by qPCR	miRs in heart tissue by microarray analyses and qPCR
[40]	2012	CVB3-infected mice: (i) susceptible to myocarditis, (ii) resistant to myocarditis	non-infected mice	miRNAome	221 deregulated miRs in susceptible mice and 78 deregulated miRs in resistant mice compared to non-infected mice	miRs in heart tissue by microarray analysis
[41]	2013	CVB3-infected mice	NC-infected mice	miR-203	↑ miR-203 in CVB3-infected mice	miRs in heart tissue by microarray analysis and qPCR
[42]	2016	VMC mice	healthy mice	miR-19b	↑ miR-19b in VMC mice	miRs in heart tissue by microarray analysis and qPCR
[43]	2012	VMC mice	non-infected mice	miR-1	↑ miR-1 in VMC mice	miRs in heart tissue by qPCR
[44]	2013	CVB3-infected mice	NC-infected mice	miRNAome	↑ miR-21, miR-29a, miR-146, miR-374 and ↓ miR-23a in VMC mice	miRs in heart tissue by microarray analysis and qPCR
[45]	2021	AMC or VMC mice	MI mice	mmu-miR-721	mmu-miR-721 present in MC mice but absent in MI mice	miRs in plasma by microarray analyses and qPCR
[46]	2020	20 CVB3-infected mice	20 non-infected mice	miR-217 miR-543	↑ miR-217 and miR-543 in study groups compared to controls	miRs in blood and heart tissue by qPCR
[47]	2018	30 VMC mice	30 healthy mice	miR-381	↓ miR-381 in VMC mice compared to healthy mice	miRs in serum and heart tissue by qPCR

↑—increased, ↓—decreased, AMC—autoimmune myocarditis, CVA2—Coxsackievirus A2, CVB3—Coxsackievirus B3, MC—myocarditis, MI—myocardial infarction, miR—microRNA, NC—negative control, qPCR—quantitative polymerase-chain-reaction, ref.—reference, RNA—ribonucleic acid, VMC—viral myocarditis.

**Table 2 biomedicines-11-02723-t002:** Summary of recent studies regarding interventions in VMC.

Ref.	Year	Population	Comparison	miRNA	Outcome	Methodology
[48]	2022	VMC mice	healthy mice	miR-324-3p	↑ miR-324-3p in VMC mice	miRs in heart tissue by qPCR VP1 by qPCR and WB
		miR-324-3p-overexpressing VMC mice	miR-324-3p-silenced VMC mice		↓ viral replication in miR-324-3p-overexpressing VMC mice	
[49]	2015	miR-221/-222-antagonist-treated CVB3-infected mice	miR-NC-treated CVB3-infected mice	miR-221 miR-222	↑ cardiac necrosis, inflammatory cells ratio, viral load, IFN-γ in miR-221/-222-inhibited mice	miRs in heart tissue by qPCR necrosis, inflammatory cells ratio by HP IFN-γ and viral load by qPCR
[50]	2013	10 miR-21-treated CVB3-infected mice	10 non-infected mice 10 miR-NC-treated CVB3-infected mice	miR-21	↓ miR-21 in VMC mice compared to healthy mice ↓ inflammation and ↑ survival, viral load in miR-21-treated VMC mice compared to controls	miRs in heart tissue by qPCR viral load by qPCR inflammation by HP
[51]	2019	miR-21-treated VMC mice	non-mammal-miR-treated VMC mice	miR-21	↓ viral load in miR-21-treated VMC mice	miRs in heart tissue by qPCR viral load by plaque assay
[52]	2022	mammal miR-Ts-CVB3 infected mice	non-mammal miR-Ts-CVB3-infected mice	miR-206 miR-29a-3p miR-124-3p	↓ viral replication in miR-206-Ts- CVB3-infected mice compared to control group	miRs in mice cells by qPCR CVB3 titers by plaque assay
[53]	2015	miR-133/206-Ts-engineered-CVB3-infected mice	NC-miR-Ts-engineered-CVB3-infected mice CVB3-infected mice	miR-133 miR-206	↓ viral replication and ↑ survival in miR-133/206-Ts-engineered-CVB3-infected mice	miRs in heart tissue by qPCR viral replication by plaque assay survival in timeslot
[54]	2021	8 VMC miR-133a-overexpressing rats	8 healthy rats 8 VMC rats 8 VMC miR-133a-silenced rats	miR-133a	↓ IL-1, IL-6, TNF-α (IFs) in VMC miR-133a-overexpressing rats compared to other VMC rats	miRs in heart tissue and CM by qPCR IFs by ELISA
[55]	2013	8 miR-21- and 8 miR-146b-inhibited VMC mice	non-infected mice 8 miR-NC-treated VMC mice 8 VMC mice	miR-21 miR-146b	↑ miR-21 and miR-146b in VMC mice compared to HCs ↓ IL-17, IL-6, (IFs), Th17-cells, inflammation, HW/BW ratio in miR-21/146b-inhibited mice compared to other VMC mice	miRs in heart tissue by microarray analysis and qPCR IFs by qPCR Th17 cells by FC inflammation by HP
[56]	2018	10 miR-21-inhibitor-treated CVB3-infected mice	10 non-infected mice 10 CVB3-infected mice 10 miR-NC-treated CVB3-infected mice	miR-21	↑ cardiac function and ↓ myocardial pathological changes in mice treated with miR-21-inhibitor compared to other CVB3-infected mice	miRs in heart tissue by qPCR cardiac function in ECHO heart morphology by HP
[57]	2019	10 miR-1/133-mimic-treated VMC mice	10 healthy mice 10 VMC mice 10 miR-NC-treated VMC mice	miR-1 miR-133	↓ miR-1, miR-133 in VMC mice compared to healthy mice ↓ EF, FS, heart damage and inflammation, cell apoptosis in the group treated with mimic compared to other VMC mice	miRs in heart tissue by qPCR EF, FS in ECHO heart morphology by HP cell apoptosis by TUNEL staining
[58]	2018	18 miR-223-treated CVB3-infected mice	10 non-infected mice 20 CVB3-infected mice 14 NC-treated CVB3-infected mice	miR-223	↓ miR-223 in VMC mice compared to non-infected mice ↓ IL-6, IFN-γ (IFs) in miR-223-treated mice compared to other CVB3-infected mice	miRs in heart tissue and heart-infiltrating macrophages by qPCR IFs by ELISA
[59]	2021	10 agomiR-425-3p-treated CVB3-infected mice	10 healthy mice 10 CVB3-infected mice 10 NC-agomiR-treated CVB3-infected mice	miR-425-3p	↓ IL-6, IL-12, TNF-α (IFs) in VMC mice treated with miR-425-3p compared to other VMC mice	miRs in heart tissue by qPCR IFs by ELISA
[60]	2016	miR-155-inhibited CVB3-infected mice	non-infected mice CVB3-infected mice	miR-155	↑ miR-155 in VMC mice compared to non-infected mice ↑ survival, FS and ↓ IL-12, TNF-α (IFs) in miR-155-inhibited VMC mice compared to other VMC mice	miRs in heart tissue by qPCR FS in ECHO IFs by ELISA
[40]	2012	miR-155-inhibited CVB3-infected mice	miR-NC-treated CVB3-infected mice	miR-155	↓ body weight loss, leukocytosis, monocyte infiltration in miR-155-inhibited mice	miRs in heart tissue by microarray analysis monocytes infiltration by HP
[61]	2014	miR-155-treated CVB3-infected mice	miR-NC-treated CVB3-infected mice	miR-155	↑ survival in miR-155-treated mice	miRs by qPCR

↑—increased, ↓—decreased, CVB3—Coxsackievirus B3, ECHO—echocardiography, EF—ejection fraction, ELISA—enzyme-linked immunosorbent assay, FC—flow cytometry, FS—fractional shortening, HP—histopathology, HW/BW—heart weight/body weight, IFN—interferon, IFs—inflammatory factors, IL—interleukin, miR—microRNA, NC—negative control, qPCR—quantitative polymerase-chain-reaction, ref.—reference, TNF—tumor necrosis factor, TUNEL—terminal deoxynucleotidyl transferase dUTP nick end labeling, VMC—viral myocarditis, VP1—viral capsid protein 1, WB—Western blotting.

**Table 3 biomedicines-11-02723-t003:** Summary of recent studies regarding interventions in autoimmune myocarditis.

Ref.	Year	Population	Comparison	miRNA	Outcome	Methodology
[62]	2021	24 antagomiR-21a-5p-treated AMC mice	24 AMC mice control group	miR-21a-5p	↓ myocardial inflammation and fibrosis in mice treated with miR-antagonist compared to the control group	miRs in heart tissue by qPCR inflammation and fibrosis by HP
[63]	2020	mice injected with exosomes from AMC mice	mice injected with exosomes from healthy mice	miR-142	↑ miR-142 in exosomes from AMC mice compared to healthy mice ↑ CD4^+^ cell infiltration in mice injected with exosomes from AMC mice	miRs in exosomes by qPCR CD4^+^ cell infiltration in mice heart tissue by HP
[64]	2020	miR-223-3p-overexpressing AMC mice	healthy mice miR-NC-treated AMC mice miR-223-3p-inhibited AMC mice	miR-223-3p	↓ miR-223-3p in AMC mice compared to healthy mice ↓ inflammation, ↑ EF in miR-223-3p-overexpressing AMC mice than in other AMC mice	miRs in serum and DCs by qPCR inflammation by HP EF in ECHO
[65]	2019	agomiR-141-3p-treated AMC mice	healthy mice miR-NC-treated AMC mice	miR-141-3p	↓ miR-141-3p in AMC mice compared to healthy mice ↓ heart hypertrophy, ↓ IFN-γ, TNF-α, IL-2, IL-6, and IL-17 (IFs) in agomiR-141-3p-treated AMC mice compared to miR-NC-treated AMC mice	miRs in heart tissue by qPCR cardiac function in ECHO IFs by ELISA
[66]	2016	antagomiR-155-treated AMC mice	healthy mice NC-treated AMC mice	miR-155	↑ miR-155 in AMC mice compared to healthy mice ↓ Th17-cells infiltration, ↑ EF, FS in antagomiR-155-treated AMC mice compared to other AMC mice	miRs in heart tissue, PBMCs, and splenic CD4^+^ T cells by qPCR Th17-cells by FC EF, FS in ECHO
[67]	2015	7 miR-590-3p-treated AMC rats	7 healthy rats 7 miR-NC-treated AMC rats	miR-590-3p	↓ miR-590-3p in AMC rats compared to healthy rats ↓ IL-6, TNFα (IFs) in miR-590-3p treated rats compared to other AMC rats	miRs in heart tissue and IFs expression by qPCR

↑—increased, ↓—decreased, AMC—autoimmune myocarditis, DCs—dendritic cells, ECHO—echocardiography, EF—ejection fraction, ELISA—enzyme-linked immunosorbent assay, FC—flow cytometry, FS—fractional shortening, HP—histopathology, IFN—interferon, IFs—inflammatory factors, IL—interleukin, miR—microRNA, NC—negative control, PBMCs—peripheral blood mononuclear cells, qPCR—quantitative polymerase-chain-reaction, ref.—reference, RNA—ribonucleic acid, TNF—tumor necrosis factor.

## Data Availability

Not applicable.

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
