# Peer review of "MicroRNAs in Myocarditis—Review of the Preclinical In Vivo Trials"

_biomedicines, 2023, doi:10.3390/biomedicines11102723_

Round 1

Reviewer 1 Report

dear authors, please:

1- you may consider to change your title as after reading the title one cannot get the nature nor the contents of your research

2-there is no separate methods section. I recommend you depict the methods more clearly and provide detailed information about your study

3-Fig 1 should be a diagram with more details, excluded studies, ...

4-are the use of pictures in Fig 1 necessary? they don't carry any information

5-Fig 2 has some pictures and a small graph (maybe Kaplan Meier, not sure) are they original? and why are they needed

6-the conclusion is so short and does not include each session conclusion

Reviewer 2 Report

Manuscript titled “ Insights from Preclinical In Vivo Trials Into the Role of MicroRNAs in Myocarditis “ is a very interesting article in the field of cardiovascular diseases and biochemistry . The overall structure is of good quality and easy to read. Methods and Results are clear and results corroborate the initial hypothesis of the authors. Figures are of sufficient quality and easy to read as well as to understand to readers. However, manuscript need some improvements, specifically in Introduction and/or Discussion. Here the points: 1. In introduction or discussion authors should explain the role of PCSK9 in myocarditis with subsequent cardiovascular events and association with some immune-related biomarkers, considering the multiple immune roles of PCSK9 ( cite doi: 10.3390/cancers15051397.) 2. In discussion authors should explain the potential cardioprotective nutraceuticals that in prclinical studies demonstrated reduction in several biomarkers associated to myocarditis, heart failure and cardiac inflammation; i.e quercetin, EPA and DHA and polydatin (cite doi: 10.3389/fonc.2021.680758). 3. Authors should add a brief description of IL-1 role in myocarditis and potential pharmacological intervention Based on these changes, the article could be suitable for publication in this journal.

Reviewer 3 Report

This review searched PubMed Database and included 30 studies regarding miRNAs in myocarditis. These 30 studies were further divided into three sections, namely dysregulated miRNAs, Interventions of miRNAs in viral myocarditis, and Interventions of miRNAs in autoimmune myocarditis. This is a systemic study with sufficient information. The following suggestions may be considered.

1. Authors searched PubMed database for inclusion of miRNAs. This selection bias may result that some more studies about miRNAs in myocarditis were missing since PubMed database did not index all the published papers. Authors need to state this limitation or explain why they only searched PubMed.

2. In the Introduction section, more information about the epidemiology of myocarditis may be provided. For example, the incidence rate and histological features of myocarditis from autopsy studies (for ref: PMID: 28122325). Also, in addition to viruses (most commonly), bacteria, or autoimmunization, medicinal drugs are also the etiology of myocarditis (such as antipsychotics, PMID: 34733639).  Authors may further extend these information in the 1.1 section.

3. It seems there is no need for Figure 2 since it does not summarize any valid information.

Round 2

Reviewer 1 Report

thank you very much for your answers.

Please also revise the abstract according to the changes 

Reviewer 3 Report

Authors addressed all my concerns.

Author Response

Dear Reviewer,

We thank you once again for taking the time to evaluate our work and for providing valuable comments.

We thank you for your time and consideration.